# Effect of Free-Range and Low-Protein Concentrated Diets on Growth Performance, Carcass Traits, and Meat Composition of Iberian Pig

**DOI:** 10.3390/ani10020273

**Published:** 2020-02-11

**Authors:** Juan F. Tejeda, Alejandro Hernández-Matamoros, Mercedes Paniagua, Elena González

**Affiliations:** 1Food Science and Technology, Escuela de Ingenierías Agrarias, Universidad de Extremadura, Avda. Adolfo Suárez s/n, 06007 Badajoz, Spain; fregepower@hotmail.com; 2Research University Institute of Agricultural Resources (INURA), Avda. de Elvas s/n, Campus Universitario, 06006 Badajoz, Spain; malena@unex.es; 3Centro de Investigaciones Científicas y Tecnológicas de Extremadura (CICYTEX-La Orden), Junta de Extremadura 06187 Guadajira, 06187 Badajoz, Spain; mercedes.paniagua@juntaex.es; 4Animal Production, Escuela de Ingenierías Agrarias, Universidad de Extremadura, Avda. Adolfo Suárez, 06007 Badajoz, Spain

**Keywords:** Iberian pig, extensive system, low-protein diet, carcass, meat quality, fatty-acid profile

## Abstract

**Simple Summary:**

It is generally assumed in the Iberian pig sector that substitution of traditional free-range rearing, with acorns and grass, by mixed diets affects intramuscular fat content and fatty-acid composition, among others, causing a decrease in meat quality. As mixed diets are usually formulated with higher protein contents than those supplied by natural resources consumed by Iberian pig fed extensively, we hypothesized that the use of a low-protein diet in the final fattening period of pig could be a suitable strategy to improve meat and dry-cured product quality. However, it is also necessary to evaluate the effect of this strategy on performance and carcass traits of pigs. In this study, we found that Iberian pigs fed on low-protein diets had higher intramuscular fat content and different meat composition compared to pigs fed on concentrates with standard protein levels, which could be a suitable way of improving the Iberian pig meat and dry-cured product quality.

**Abstract:**

The feeding system is one of the main factors influencing the Iberian pig meat quality. This experiment was undertaken to evaluate the influence of feeding diets containing different levels of protein on performance, carcass, and meat quality of Iberian pigs. To that aim, 24 castrated male Retinto Iberian pigs with an average weight of 116 kg were fed under free-range conditions with acorns and grass (FR), and on concentrated diets in confinement with standard (SP) and low protein content (LP). The crude protein content in acorns was lower than that in the grass and SP diet, but similar to that in the LP diet. FR pigs needed more time to achieve slaughter weight than LP and SP pigs. Iberian pigs fed on low-protein diet (FR and LP) had a higher intramuscular fat content in the musculus serratus ventralis than SP pigs. The influence of diet on the fatty-acid composition was reflected more markedly in subcutaneous fat than in muscles. FR pigs showed a higher level of C18:1 n-9 and total polyunsaturated fatty acids and lower total saturated fatty acids in subcutaneous fat than LP and SP. It is concluded that diets with low protein levels do not affect Iberian pig productive traits but change the meat composition, rendering them an interesting strategy to improve the quality of Iberian pig meat and dry-cured products.

## 1. Introduction

The Iberian pig is an autochthonous breed from the southwest Iberian Peninsula, characterized by its high-quality meat and dry-cured products (mainly hams, shoulders, and loins) [1]. This high quality is the consequence of several factors such as genetics, crossbreeding, rearing system, and processing conditions. Within the factors included under the rearing system, feeding seems to be the key one influencing Iberian product quality [2]. Originally, Iberian pigs were reared under free-range conditions in the *dehesa*, a Mediterranean forest system, based on natural resources, mainly fallen acorns and pasture, playing an important role in the agricultural and pastoral systems and, therefore, in the economy of these rural areas [3]. Unfortunately, an extensive traditional feeding production regime is not always feasible because the availability of natural resources is limited. In addition, an increase in demand for both fresh and dry-cured products from Iberian pigs currently involves the use of conventional mixed diets to produce a high proportion of Iberian pigs under intensive conditions. However, the replacement of free-range rearing and substitution of the natural feed by conventional mixed diets in Iberian pigs produce a markedly decrease in the sensory attributes of dry-cured products and, consequently, a lower acceptability [4]. It is well known that nutritional strategies are the main influential factors in meat quality of pigs. The acorn is an energy food rich in fat and carbohydrates; however, it has very low protein content and its amino-acid profile indicates that lysine is the main limiting amino acid [3]. On the other hand, although protein from pasture may be important to overcome the shortage in acorns, it is not enough to cover dietary requirements [5]. To increase pig chain sustainability, it is necessary to optimize the nutrient efficiency, since it may reduce nutrient excretion and production costs, making the reduction of the dietary protein content a priority objective in pig production [6,7]. In this sense, previous studies into intensive nutritional management were carried out to determine the influence of diets that differed widely in protein/energy ratio on performance, carcass, and meat quality traits of Iberian pig [8,9]. It is well known than the reduction of some nutrients, such as protein, could increase fat deposition in pigs [10,11]. With regard to Iberian pig, a high amount of intramuscular fat (IMF) was highlighted as one of the most relevant aspects of meat quality [12]. Thus, increasing the IMF levels is generally considered as a way of enhancing the quality of fresh pork and dry-cured products [12], and a low-protein diet could be considered as a strategy to increase fat deposition, allowing not only a decrease in feed costs, but also an increase in sustainability of pork production.

Thus, as free-range rearing of Iberian pigs, with a feed characterized by a high energy density and a low amount and quality of proteins, increases the IMF [13] and modifies the color of meat and dry-cured products [14], we hypothesized that feeding Iberian pigs with a low-protein diet during the final fattening period prior to slaughter under intensive conditions could be an adequate strategy to improve the meat quality. In addition, another purpose of this work was to study the influence of this type of a protein-restricted diet on the performance and carcass traits of Iberian pigs.

## 2. Materials and Methods

### 2.1. Animals and Diets

This study was carried out with 24 castrated Iberian male pigs of the Retinto variety. This variety belongs to the line Valdesequera (Extremadura Government, Badajoz, Spain), and it is recognized in Spain’s official Iberian herd book (Spanish Association of Iberian Purebred Pig Breeders, AECERIBER). During the period prior to the experiment, the animals were kept from birth to the beginning of the fattening phase under intensive rearing conditions. Animals began the fattening phase with an average initial live weight of 116.0 kg (pooled SD = 5.9 kg) and age of 425 days. The pigs were divided into three groups (*n* = 8) according to feeding regime. One group was reared free range (FR), according to the traditional way, in which pigs are fed on natural resources, mainly acorns (*Quercus rotundifolia*) and grass, from November to January. The other two groups of pigs were raised on the experimental farm in confinement (housed outdoors, 230 m^2^/pig) and offered two different experimental diets, formulated to the same metabolizable energy (ME) value, 4100 kcal/kg dry matter (DM) (according to Fundación Española para el Desarrollo de la Nutrición Animal (FEDNA) [15]); one group was fed on a standard protein diet (SP) and the other one on a low-protein diet (LP). Standard protein diet was formulated to meet all the nutrient requirements for fattening the Iberian pigs; however, LP diet was designed as a protein-deficient diet, in order to obtain a low protein intake that simulates feeding of Iberian pigs reared free range. The main ingredients of the two diets were maize (150 g/kg), maize starch (400 g/kg), alfalfa meal (100 g/kg), and high-oleic sunflower oil (60 g/kg). The LP diet was also formulated with wheat (252 g/kg), resulting in a concentrated feed with 6.6 g/100 g DM of crude protein and 0.2 g/100 g DM of lysine, and the SP diet with wheat (55 g/kg) and soybean meal (44% crude protein), resulting in a concentrated feed with 12.8 g/100 g DM and 0.7 g/100 g DM of lysine. The diets contained also bicalcium phosphate (20 g/kg), sodium chloride (5 g/kg), vitamin and mineral premix (3 g/kg), and binder (10 g/kg). The chemical composition of diets (acorns, grass, and concentrated feed) was determined according to standard methods [16]: moisture (reference 935.29), crude protein (reference 954.01), crude fat (reference 920.39), crude fiber (reference 962.09), and ash (reference 942.05). The fatty-acid composition of diets was assayed by gas chromatography after lipid extraction according to the Bligh and Dyer [17] method and acidic transesterification [18].

### 2.2. Handling, Slaughtering, and Carcass Traits

Pigs from the three groups studied, FR, LP, and SP, had free access to water and feed throughout the trial. To achieve ad libitum feeding of pigs raised in confinement (LP and SP), a sufficient amount of feed was weighed and added to feed hoppers manually once a day, ensuring that feed was always available to the animals. Feed consumption was recorded weekly. Daily feed intake was calculated by weighing the feed leftovers, which were determined by weighing the entire feeder (the entire feeder included the feed hoppers) and subtracting the weight of the empty feeder. Pigs on the LP diet had a daily feed intake of 5161 g DM/pig versus 5088 g DM/pig in the SP group. As FR pigs were fed in freedom, with the resources that nature provides, it was not possible to measure their feed intake directly, but daily acorn pulp and grass intakes of 3600 and 655 g DM/pig, respectively, were estimated according to the previous studies published in Iberian pig (see Appendix A) [3,19,20]. Pigs were weighed every week from the beginning of the fattening period to slaughter. The selection criterion for the slaughter of the pigs was weight increase. When the average weight of the batch increased by about 58 kg, all the pigs of a batch were transported to the abattoir. All the pigs were weighed unfasted 24 h before slaughter, and these weights were used to determine both final weight (174.2 kg and SD = 6.1) and carcass performance. Feed was withheld from animals for 12 h before slaughtering. FR pigs were locked in a pen and fed with acorns collected from the same place where they usually ate, until the pre-slaughter fasting. Pigs were slaughtered by electrical stunning and killed by exsanguination. Then, they were scalded, skinned, eviscerated, and split down both sides of the vertebral column according to the standard commercial procedures of the Iberian pig industry. Hot carcass weights without pelvic renal fat were recorded and used to calculate carcass yield. Hams and shoulders were removed from the carcasses and weighed 2 h postmortem. Serratus ventralis (SV) and longissimus thoracis and lumborum (LTL) muscles were also dissected from the carcasses, trimmed of external fat, and weighed within 45 min after slaughter. The weights of untrimmed hams and shoulders were recorded 2 h postmortem.

### 2.3. Meat Quality Traits and Fatty-Acid Profile of Samples

After collection of carcass data, a sample of subcutaneous adipose tissue at the level of the tailbone was chosen for lipid analysis. Then, a sample of approximately 300 g in the middle part of the SV and another one from the musculus longissimus lumborum (LL) was excised from all pigs. Meat samples were stored for chemical analysis in individual plastic bags and vacuum-packaged at −20 °C until subsequent analyses. The LL and SV samples were thawed inside the vacuum-packaged bags for 24 h at 4 °C, removed from packages, sliced, and exposed to light for 30 min before color measurement. The following color coordinates were determined: lightness (L*), redness (a*, red to green), and yellowness (b*, yellow to blue), according to CIELAB color space [21]. The color parameters were determined using a Minolta CR-300 colorimeter reflectance spectrophotometer (Minolta Camera Co., Osaka, Japan). Before use, the colorimeter was standardized using a white tile (mod CR-A43) using illuminant D65, 0° standard observer, and an 8 mm port/viewing area. The measurements were repeated at three randomly selected locations on each LL and SV slice and averaged for statistical analysis. The pH at 24 h of LL and SV samples was measured using a pH meter specific for meat products (model HI 99163, HANNA, Smithfield, RI, USA). Moisture (oven air-drying method), protein (Kjeldahl nitrogen), and ash (muffle furnace) were analyzed following official methods [16]. Lipids from subcutaneous fat samples were extracted in a microwave oven following the method described by De Pedro et al. [22]. IMF was extracted and quantified according to the method described by Bligh and Dyer [17]. Fatty-acid methyl esters from the lipids obtained were prepared by acidic transesterification in the presence of sodium metal (0,1 N) and sulfuric acid (5% sulfuric acid in methanol) [18], and they were analyzed by gas chromatography, using a Hewlett-Packard HP-4890 Series II gas chromatograph equipped with a split/splitless injector and a flame ionization detector (FID). Separation was carried out on a polyethylene glycol capillary column (30 m long, 0.25 mm inner diameter (id), 0.25 μm film thickness) (HP-INNOWax) maintained at 260 °C for 25 min. Injector and detector temperatures were 320 °C. The carrier gas was nitrogen at 1.8 mL/min. Individual fatty acids were identified by comparison of their retention times with those of reference standard mixtures (Sigma Chemical Co., St. Louis, MO, USA). Results were expressed as the percentage of total fatty acids present, considering a total of 16 fatty acids, none less than 0.1%.

### 2.4. Statistical Analysis

For descriptive data analysis, the mean and the standard error of the mean were used. The pig was used as the experimental unit. Significance of difference (*p* < 0.05) between dietary treatments was determined by one-way ANOVA followed by Tukey multiple comparison test. The general linear model procedure of the SPSS package (SPSS for Windows Ver. 19.0; SPSS Inc., Chicago, IL, USA, 2004) was used. 

## 3. Results

### 3.1. Diets

Table 1 shows analysis of the chemical and fatty-acid composition of experimental diets, acorns, and grass. Acorns, the main component of FR Iberian pig diets, contain lower crude protein content (5.2% DM) than grass (19.1% DM) and the SP diet (12.8% DM), but similar content to the LP diet (6.6% DM). Acorns presented higher levels of fat (7.9% DM) than grass (4.6% DM), but similar levels to those found in LP and SP diets (8.3% and 8.0% DM, respectively). Acorns, LP, and SP diets exhibited higher proportions of oleic acid (64.0%, 72.5%, and 71.9%, respectively) than grass (13.2%). However, grass presented higher levels of linolenic acid (37.5%) compared to acorns, LP, and SP diets (1.0%, 0.6%, and 0.9%, respectively).

### 3.2. Pig Performance and Carcass Traits

Growth performance and carcass traits are shown in Table 2. Pigs from the FR group grew slower (1.05 kg/day) and needed more time (56.8 days) to get to the slaughter weight than pigs from LP and SP groups, which had significantly higher growth rates (1.27 and 1.24 kg/day, respectively) and only needed 45.4 and 46.3 days, respectively, to get to the slaughter weight. No significant differences were found between LP and SP groups in the average daily gain (ADG) and in the number of days to get to the final weight. Carcass weight and carcass yield were not influenced by production system or feed type. With regard to the cutting of pigs, the largest differences were in loin weight and yield. The three groups showed significant differences for LTL weight, with the highest value for SP (2.41 kg), followed by LP and FR (2.16 and 1.90 kg, respectively). Although SV showed the same behavior as LTL, statistically significant differences were not found. With respect to ham and shoulder weights, no differences were found among groups.

### 3.3. Meat Composition

The effect of feeding diet on proximate chemical composition, instrumental color coordinates, and pH of LL and SV is presented in Table 3. Significant differences between feeding diets were observed in SV muscle. SV from pigs fed on low levels of protein (FR and LP groups) had a higher IMF content (7.95 and 7.72 g/100 g, respectively) than pigs fed on standard levels of protein (SP group) (6.23 g/100 g). There were decreased pH levels measured at 24 h in FR pigs (5.73), as opposed to SP pigs (5.99), with the LP group showing intermediate values (5.87). With respect to color of SV, FR pigs had higher values of L* (39.53) than LP and SP (36.71 and 36.18, respectively). In LL muscle, the same trend was observed as in SV; however, the differences between feeding diets were not significant.

The influence of feeding diets on the fatty-acid composition of subcutaneous backfat is shown in Table 4. There were significant differences between Iberian pigs fed in free-range rearing conditions (FR) and pigs fed in confinement with experimental concentrated diets (LP and SP). FR pigs showed a higher percentage of C18:1 n-9, C18:2 n-6, C18:3 n-3, C20:4 n-6, and total polyunsaturated fatty acids (PUFA), and lower percentages of C16:0, C18:0, C20:0, and total saturated fatty acids (SFA) than LP and SP groups.

Considering the influence of diet on the fatty-acid composition of LL and SV (Table 5), the only significant differences were observed in the total PUFA of SV, mainly due to the differences in C18:2 n-6, C18:3 n-3, C20:2 n-9, and C20:3 n-3.

## 4. Discussion

### 4.1. Experimental Diets

The chemical composition of the acorns and grass in this work showed similar values to those previously published [20,23]. These results show that the protein content in acorns consumed by Iberian pigs during the fattening period in free-range conditions is low compared to diets used in lighter pig systems. Moreover, acorn protein content is constrained by an unbalanced amino-acid profile, with lysine as the main limiting amino acid, according to Nieto et al. [8], who reported average values of 0.2 g lysine/100 g DM in the acorn kernel (Iberian pigs remove the acorn hull to ingest only the kernel). Even though the provision of supplementary protein via pasture could cause an increase in protein deposition in pigs, García-Valverde et al. [3] reported that the amount of protein supplied by pasture is not enough to supply the protein needed during the fattening phase in a free-range system. Thus, in our study, the experimental SP diet was designed to supply the total daily needs of Iberian pig in the fattening phase according to García-Valverde et al. [5], who stated that the maximum potential for the deposition of lean tissue in Iberian pigs during the fattening period is attained when the pigs are fed with a diet which provides 9.5 g crude ideal protein/100 g DM and 0.7 g lysine/100 g DM. On the other hand, the LP diet was designed to simulate, under controlled conditions, the nutrients received by the Iberian pigs in free-range conditions. For this, we took into account both the composition and the proportion of acorns and grass consumed by the pigs according to the studies of Rodríguez-Estévez et al. [24]. Total nutrient intake also depends on the amount of food consumed by the pigs in each treatment. In our study, confined pigs (SP and LP) had very high feed intake, far above that described by García Valverde et al. [5] and closer to that described by Dunker et al. [25] after a restricted feeding period. With respect to acorn and grass, the intakes estimated in our work (3.60 kg DM/day and 0.65 kg DM/day of acorn kernel and grass, respectively) are higher than (2.9 kg DM/day and 0.5 kg DM/day) [20] or similar to (3.6 kg DM/day and 0.38 kg DM/day) [24] those previously reported. Although the protein/energy ratio in LP pigs was much lower than recommended by García-Valverde et al. [5], the daily protein intake was not as low, due to the high daily feed intake. Thus, the estimated daily protein intake in FR pigs was 312 g crude protein/day (187 g from acorn and 125 g from grass), while, in LP and SP pigs, it was 340 and 651 g crude protein/day, respectively. Nevertheless, although the total protein intake between LP and FR could be similar, there may be a difference in the amount of available protein, due to the low protein retained/protein intake ratio in acorns and pasture (0.078 and 0.202, respectively) [3] and in concentrated feed (0.212) [5].

### 4.2. Pig Performance and Carcass Quality Traits

With respect to productive parameters, ADGs in our study were in general higher than those found in Iberian pigs fed during the fattening period on formulated diets in a confinement system [5,26] or fed with acorns and grass in a free-range system [27]. This could be explained by the pigs in our experiment being older at the beginning of the fattening period, as previously demonstrated [28], probably due to compensatory growth as a result of previous food restriction [27]. A significantly lower ADG was observed in FR compared to SP and LP pigs (*p* < 0.05). Several studies evidenced differences in ADG between free-range reared Iberian pigs and those raised in confinement, due to the effect of physical activity [29] and climatic conditions [30]. However, all pigs in this work had the same thermoregulation needs, given that SP and LP pigs were outdoors and near to those from the FR group. Thus, most of the differences could be due to the expenditure of energy for displacement. Although SP and LP pigs were confined outdoors within a large plot of 230 m^2^/pig, they did not have to move to search for food; hence, this cost was higher in the FR animals. No dietary effect on ADG between LP and SP groups was observed, in accordance with previous studies in Alentejano (Iberian) pigs [31] and in heavy pigs [32,33,34,35]. In contrast, other authors found that feeding pigs ad libitum with protein- or lysine-deficient but adequate-energy diets during the finishing phase reduces ADG rate [36,37,38]. Differences between the abovementioned studies could be explained by the different growth rates of the pig breed [6], the pig body weight when the protein restriction is carried out [38], and the deficiencies in protein and essential amino-acid levels [31].

No differences in carcass yield were found in our work, in accordance with previous results in Iberian pigs [26] and in other pig breeds [35,39]. In contrast, Rey et al. [40] found higher carcass yield in pigs fed in confinement than in those fed under free-range conditions with acorn and grass, probably due to the greater fiber content in grass compared to concentrate diet, which could increase the development of the digestive system (mainly large intestine), as evidenced Roskosz et al. [41] in wild pigs fed on diets with a high cellulose content. However, the higher feed intake of intensively reared pigs increased the gut fill [19], which could compensate for the greater development of the digestive system from free range-reared pigs. Related to dietary protein content, our results are in agreement with previous papers [35,42], indicating that it is possible to reduce dietary crude protein without affecting growth performance and carcass composition as long as daily amino-acid supplies are adequate [34]. Only in loins were significantly lower levels of weight and yield detected in FR than in SP pigs, with intermediate values in LP. The effect of a different protein/energy ratio in the three diets studied could be more significant in loins compared to other cuts, such as ham or shoulder, due to loin being a leaner cut, and the reduction in the proportion of protein relative to energy in the diet consistently increases fat deposition and decreases muscle synthesis [43].

### 4.3. Meat Quality

The higher IMF content of Iberian pigs reared outdoors compared to those reared in intensive conditions and fed on concentrated diets is well known by farmers and dry-cured ham producers [44] and was previously reported [13,23]. This could be due to the high intake during the fattening period prior to slaughter of acorns which have a high caloric value (Rodríguez-Estévez et al. [20] estimated a daily feed intake of 2.92 kg DM acorn) and low protein content with an unbalanced amino-acid profile [8]. In the current study, there were no differences in IMF between FR pigs and LP pigs, but there were differences between both previous and SP pigs in SV. These results could be explained by the low protein content of the LP diet, which was similar to that of FR, and lower than that for the SP treatment. When insufficient dietary protein content is provided to pigs, excess energy is diverted to fat deposition [6]. Moreover, evidence suggests that, with low-protein diets, lipogenic enzymes are expressed more readily in muscle than in subcutaneous fat [45]. Therefore, diet composition, particularly the protein/energy ratio, can be used to increase fatness, with a consequent effect on performance [46]. Indeed, feeding pigs ad libitum with protein- or lysine-deficient but adequate-energy diets during the growing or finishing phases was shown to increase IMF proportion [36], which corroborates our results. In this sense, Schiavon et al. [47] found higher fat cover and thickness and marbling in hams from heavy pigs fed low-protein diets. More recently, Li et al. [38] evidenced the effects of low-protein diets on variations in the expression of two genes (*ACC* (acetyl-CoA carboxylase alpha) and *HSL* (hormone-sensitive lipase)) related to lipid metabolism, thereby promoting fat deposition in the muscle, which agrees with the results of the present study. The effect of feeding diets on chemical meat composition was only observed in SV. IMF content was lower in LL than in SV, which could be related to the type of muscle metabolism, as it is generally accepted that a higher proportion of oxidative fibers implies a greater IMF content [44].

With respect to color, the higher luminosity (L*) values in SV from pigs reared in the FR system than in that from pigs reared in intensive conditions (LP and SP) could be attributed to the combined effect of feed characteristics and the environment, and not only to the exercise of pigs, as demonstrated by López-Bote et al. [48] in studies with Iberian pigs. Thus, the higher L* value of SV from FR compared to LP and SP pigs could also be related to the higher IMF content in FR animals, in accordance with the results of Andrés et al. [44], who found a positive relationship between L* value and fat content in Iberian pork. In the same way, Tejerina et al. [14] found that pigs reared in extensive conditions with acorns and grass had higher L* and b* values in LL and SV than those from Iberian pigs raised in intensive conditions with concentrated diets. In our study, LL showed a same tendency as SV, albeit without significant differences, which could be related to the lower IMF content in LL than in SV, in accordance with Tejerina et al. [14].

### 4.4. Fatty-Acid Composition

It is well known that the fatty-acid composition of pig tissues is affected principally by the fatty-acid composition of feed [49]. In our study, the influence of feeding background on the fatty-acid composition of porcine tissues was reflected more markedly in subcutaneous fat than in LL and SV muscles. Even though pigs fed in confinement (SP and LP diets) had monounsaturated fatty-acid-enriched diets, a significant influence of FR on the four major fatty acids (C18:1 n-9, C16:0, C18:0, and C18:2 n-6) and SFA and PUFA of subcutaneous fat was found. FR pigs showed a higher C18:1 n-9 content at the level of the tailbone than SP and LP, reflecting the high concentration of oleic acid from acorns, in agreement with the results previously reported [50]. However, Ventanas et al. [13], studying the effect of extensive feeding vs oleic acid-enriched mixed diets in Iberian pigs, did not detect any effect on C18:1 n-9. Moreover, in our study, FR pigs exhibited a lower SFA and higher PUFA content than LP and SP, in accordance with previous studies comparing Iberian pigs fed on free-range and concentrated diets [23]. Nevertheless, the level of protein in the diet did not affect the fatty-acid composition of subcutaneous fat and LL. The greatest effect of the level of protein in diets was reflected in SV, with the biggest change being in C18:2 n-6, and subsequently total PUFA, with the LP regime exhibiting lower proportions than both FR and SP regimes. These results are in agreement with Wood et al. [6], who reported that C18:2 n-6 and C18:1 n-9 are the fatty acids whose concentrations are most affected by a reduction of protein in pig diets. Previous works showed that these two fatty acids are those most affected by changes in total fat deposition in pigs and other animal species [51]. Oleic acid is the main product of de novo fat synthesis in the pig, and it is logical that its concentration increases as the pig gets fatter. The linoleic acid obtained from the diet is then progressively diluted as fat synthesis increases, which explains the declining concentration of this fatty acid. Nevertheless, no significant effect of protein level in the diet was detected for C18:1 n-9 content in our study. The explanation of this aspect could be based on the fatty-acid composition of concentrated feed used in our study, which was rich in oleic acid. Hence, dietary protein content (and, more specifically, the protein/energy ratio) may be used to modify the degree of carcass fat and fatty-acid composition. It is generally accepted that the high sensory quality of Iberian pig meat products from pigs fed under extensive conditions when compared to those fed on mixed feeds is attributed to variation in the content and fatty-acid composition of intramuscular lipids [52,53].

## 5. Conclusions

The results of this trial show that low-protein diets during the final fattening period prior to slaughter, similar to a feeding regime in free-range conditions with acorns and grass, do not affect Iberian pig productive traits. Additionally, they increase IMF content, which is one of the quality parameters most appreciated by consumers. In this sense, there is growing interest in muscle tissue recently, owing to the effect of increasing IMF on meat quality. Thus, feeding Iberian pigs with LP diets should be an interesting strategy to improve the quality of Iberian pig meat and dry-cured products.

## Figures and Tables

**Table 1 animals-10-00273-t001:** Proximate composition (% dry matter, except for dry matter (%)) and fatty-acid profile (%) of the experimental diets (low-protein diet and standard protein diet), acorn, and grass.

Items	Diets			
Chemical Composition	LP diet	SP Diet	Acorn	Grass
Dry matter (DM)	91.6	91.3	60.5	19.4
Crude protein	6.6	12.8	5.2	19.1
Crude fat	8.3	8.0	7.9	4.6
Crude fibre	4.0	4.7	2.3	22.0
Ash	5.0	5.2	1.6	11.6
Free-nitrogen extractives	75.9	68.6	83.0	42.7
Lysine	0.2	0.7		
Fatty acids ^1^				
Palmitic acid (C16:0)	7.3	7.0	13.5	26.1
Stearic acid (C18:0)	2.9	2.7	3.3	6.1
Oleic acid (C18:1n-9)	72.5	71.9	64.0	13.2
Linoleic acid (C18:2n-6)	14.6	15.9	16.7	12.6
Linolenic acid (C18:3n-3)	0.6	0.9	1.0	37.5

LP diet = low-protein diet; SP diet = standard protein diet. ^1^ Of a total of 16 fatty acids, none less than 0.1%.

**Table 2 animals-10-00273-t002:** Productive and carcass traits (kg) and yields (%) from Iberian pigs fed the experimental diets.

Productive and Carcass Traits	FR	LP	SP	SEM	*p*-Value
Initial weight	115.6	116.3	116.1	1.213	0.975
Final weight	175.2	174.0	173.5	1.243	0.859
Weight gain	59.6	57.7	57.4	0.734	0.445
Days	56.8 ^a^	45.4 ^b^	46.3 ^b^	2.137	0.023
Carcass weight	137.2	137.3	135.2	1.169	0.721
ADG	1.05 ^a^	1.27 ^b^	1.24 ^b^	0.062	0.049
LTL weight	1.90 ^a^	2.16 ^b^	2.41 ^c^	0.054	0.000
SV weight	0.60	0.70	0.69	0.019	0.053
Ham weight	14.53	14.71	14.48	0.170	0.812
Shoulder weight	10.66	11.03	10.92	0.117	0.335
Carcass yield	78.35	78.97	77.91	0.344	0.558
LTL yield	2.82 ^a^	3.19 ^b^	3.66 ^c^	0.084	0.000
Ham yield	21.15	21.40	21.44	0.213	0.894
Shoulder yield	15.44	15.96	16.11	0.129	0.063

^a,b,c^ Values within a row with different superscripts differ significantly at *p* < 0.05. SEM, standard error of the mean. ADG, average daily gain. LTL, musculus longissimus thoracis and lumborum. SV, musculus serratus ventralis. Diets: FR, Iberian pigs reared in free-range conditions; LP, Iberian pig fed on experimental low-protein diet; SP, Iberian pig fed on experimental standard protein diet.

**Table 3 animals-10-00273-t003:** Chemical composition (g/100 g of muscle), pH, and color of musculus longissimus lumborum and musculus serratus ventralis from Iberian pigs fed the experimental diets.

Items	Musculus Longissimus Lumborum					Musculus Serratus Ventralis				
	FR	LP	SP	SEM	*p*-Value	FR	LP	SP	SEM	*p*-Value
Moisture	69.56	70.15	71.39	0.362	0.104	71.17 ^a^	71.78 ^a,b^	72.77 ^b^	0.242	0.016
Protein	22.44	21.92	22.16	0.143	0.350	19.54	19.15	19.61	0.129	0.296
IMF	6.60	6.57	5.06	0.363	0.141	7.95 ^a^	7.72 ^a^	6.23 ^b^	0.298	0.030
Ash	1.10	1.05	1.09	0.011	0.228	1.03 ^a^	1.06 ^a,b^	1.09 ^b^	0.009	0.037
pH 24 h	5.56	5.69	5.63	0.036	0.326	5.73 ^a^	5.87 ^a,b^	5.99 ^b^	0.040	0.024
L*	43.22	40.41	41.39	0.559	0.111	39.53 ^a^	36.71 ^b^	36.18 ^b^	0.563	0.025
a*	8.56	8.97	8.70	0.243	0.806	15.20	14.84	15.29	0.283	0.807
b*	6.86	6.84	6.83	0.190	0.999	9.61	8.89	9.22	0.209	0.388

^a,b^ Values within a row with different superscripts differ significantly at *p* < 0.05. SEM, standard error of the mean. IMF, intramuscular fat. Diets: FR, Iberian pigs reared in free-range conditions; LP, Iberian pig fed on experimental low-protein diet; SP, Iberian pig fed on experimental standard protein diet. L*: lightness; a*: redness; b*: yellowness.

**Table 4 animals-10-00273-t004:** Fatty-acid composition (%) of the subcutaneous fat from Iberian pigs fed the experimental diets.

Items	Subcutaneous Fat				
	FR	LP	SP	SEM	*p*-Value
C14:0	1.24	1.20	1.19	0.016	0.504
C16:0	19.60 ^a^	20.77 ^b^	20.51 ^b^	0.163	0.004
C16:1	2.19	2.17	2.31	0.075	0.728
C17:0	0.30	0.26	0.29	0.010	0.227
C17:1	0.35	0.31	0.34	0.011	0.353
C18:0	8.58 ^a^	10.36 ^b^	9.94 ^a,b^	0.264	0.009
C18:1 n-9	53.99 ^a^	52.59 ^b^	52.55 ^b^	0.235	0.010
C18:2 n-6	9.95 ^a^	8.43 ^b^	8.93 ^b^	0.195	0.002
C18:3 n-3	0.75 ^a^	0.64 ^b^	0.66 ^b^	0.018	0.014
C20:0	0.16 ^a^	0.20 ^b^	0.20 ^b^	0.006	0.019
C20:1 n-9	1.69	1.91	1.87	0.045	0.080
C20:2 n-9	0.70	0.68	0.72	0.015	0.543
C20:4 n-6	0.14 ^a^	0.12 ^b^	0.13 ^a,b^	0.003	0.025
C20:3 n-3	0.27	0.26	0.27	0.008	0.896
SFA	29.89 ^a^	32.79 ^b^	32.13 ^b^	0.404	0.004
MUFA	58.21	56.98	57.07	0.257	0.088
PUFA	11.90 ^a^	10.22 ^b^	10.80 ^b^	0.219	0.002

^a,b^ Values within a row with different superscripts differ significantly at *p* < 0.05. SEM, standard error of the mean. SFA, total saturated fatty acids; MUFA, total monounsaturated fatty acids; PUFA, total polyunsaturated fatty acids. Diets: FR, Iberian pigs reared in free-range conditions; LP, Iberian pig fed on experimental low-protein diet; SP, Iberian pig fed on experimental standard protein diet. Results are expressed as means in percentage of a total of 16 fatty acids, none less than 0.1%.

**Table 5 animals-10-00273-t005:** Fatty-acid composition (%) of the musculus longissimus lumborum and musculus serratus ventralis from Iberian pigs fed the experimental diets.

Items	Musculus Longissimus Lumborum					Musculus Serratus Ventralis				
	FR	LP	SP	SEM	*p*-Value	FR	LP	SP	SEM	*p*-Value
C14:0	1.31	1.30	1.40	0.037	0.492	1.27	1.17	1.22	0.018	0.105
C16:0	24.19	24.48	23.46	0.386	0.562	23.47	24.25	23.24	0.262	0.266
C16:1	4.66	4.15	4.89	0.133	0.058	4.04	3.51	4.08	0.139	0.182
C17:0	0.14 ^a^	0.12 ^b^	0.16 ^a^	0.007	0.044	0.19 ^a^	0.15 ^b^	0.19 ^a^	0.006	0.011
C17:1	0.20	0.16	0.21	0.010	0.119	0.25 ^a^	0.19 ^b^	0.24 ^a^	0.010	0.034
C18:0	10.22	11.41	10.38	0.271	0.152	10.24	12.21	10.40	0.376	0.053
C18:1 n-9	51.39	50.81	51.47	0.428	0.802	49.74	49.23	50.09	0.400	0.699
C18:2 n-6	5.41	4.96	5.33	0.179	0.576	7.88 ^a^	6.39 ^b^	7.52 ^a^	0.210	0.006
C18:3 n-3	0.37 ^a,b^	0.35 ^a^	0.41 ^b^	0.009	0.035	0.48 ^a^	0.35 ^b^	0.46 ^a^	0.018	0.003
C20:0	0.17	0.18	0.17	0.005	0.529	0.17	0.20	0.17	0.008	0.094
C20:1 n-9	0.87	0.91	0.85	0.026	0.596	0.96	1.14	1.05	0.035	0.118
C20:2 n-9	0.21	0.19	0.22	0.007	0.364	0.30 ^a^	0.25 ^b^	0.31 ^a^	0.008	0.002
C20:3 n-6	0.10	0.11	0.13	0.006	0.181	0.13	0.12	0.14	0.005	0.258
C20:4 n-6	0.67	0.78	0.82	0.046	0.400	0.79	0.72	0.76	0.032	0.708
C20:3 n-3	0.09	0.09	0.10	0.004	0.425	0.12 ^a^	0.09 ^b^	0.12 ^a^	0.005	0.017
SFA	36.02	37.49	35.58	0.649	0.473	35.33	37.99	35.22	0.619	0.116
MUFA	57.13	56.03	57.42	0.521	0.538	54.98	54.08	55.47	0.507	0.546
PUFA	6.85	6.48	7.00	0.237	0.673	9.68 ^a^	7.93 ^b^	9.31 ^a^	0.258	0.007

^a,b^ Values within a row with different superscripts differ significantly at *p* < 0.05. SEM, standard error of the mean. SFA, total saturated fatty acids; MUFA, total monounsaturated fatty acids; PUFA, total polyunsaturated fatty acids. Diets: FR, Iberian pigs reared in free-range conditions; LP, Iberian pig fed on experimental low-protein diet; SP, Iberian pig fed on experimental standard protein diet. Results are expressed as means in percentage of a total of 16 fatty acids, none less than 0.1%.

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
