# Peer review of "Effect of Free-Range and Low-Protein Concentrated Diets on Growth Performance, Carcass Traits, and Meat Composition of Iberian Pig"

_animals, 2020, doi:10.3390/ani10020273_

Round 1
Reviewer 1 Report
Presented manuscript is valuable, especially from practical point of wiev.
It clearly stated that low-protein diet can be applied to obtain similar results as free-range diet. What is most important, that ADG and number of days to slaughter in case of low-protein diet is similar to standard diet what is not associated with additional cost for longer rearing of pigs on the farm.
I have found some editorial errors that can be corrected:
1) First sentence in Abstract should be reworded
2) Word "retinto" should be standarized (italics vs normal - line 32 vs 83)
3) Each shortcut of word muscle (m.) should be italicized because latin name "musculus" - lines 37, 133, 191, 204-205 (incl. Table 3.), 227-228 (incl. Table 4.)
4) Similar to point 3) latin names of muscle should be italicized - lines 127-128, 204-205
5) Additional word to italicize: ad libitum (281), de novo (353), Quercus...(91)
6) Line 36 - "fed on low-protein"
7) Line 58 "markedly"
8) Line 71 - "aspects of meat quality)
9) Explain please, shortcuts in lines 93-94 - ME, DM, FEDNA
10) Explain please shortcuts below Table 3. - L*, A*, b* (i think word CIE is not necessary in each case)
11) Table 4. - "Subcutaneus"
12) Line 249 - "to try simulate"
13) Add the information about daily protein intake for SP diet (260-262)
14) Line 317 - please mention the name of genes in brackets
Reviewer 2 Report
The article “Effect of free-range and low-protein concentrated diets on growth performance, carcass traits and meat composition of Iberian pig” addresses a contemporary topic in pig nutrition, having dietary protein content several implications from the economic point of view and for environmental sustainability. Local pig breeds still lack of research on feeding managements, especially related to feeding strategies that rely on changing diet components and composition during different stages of growth. Also, feeding management is of central importance in pig production due to the influence which has on product quality. The manuscript gives interesting information of different feeding managements applied on an important local pork chain, the Iberian one. The experiment is well-designed, and results are consistent and clear reported and discussed. However, some minor revisions should be addressed before its publication on Animals. Please find below some points that, in my opinion, need to be clarified or modified:
INTRODUCTION
L77: Feed with
L83: 24 Castrated Iberian male pigs
MATERIAL AND METHODS
L93: To make the manuscript easier to read and compare with similar researches I would suggest to express the energy as DE and to use kcal or MJ
L111: The adopted feeding strategy was ad libitum, is that correct? How did you manage this? By refilling the troughs every time they were empty, or did you use an automatic system providing feed? Moreover, how did you consider any feed leftovers when or if present in order to calculate daily feed consumption?
L127-128: How was SV and LTL yield calculated? Was it in relation to slaughter weight or carcass weight? Were they trimmed before weighting? Was Subcutaneous fat totally removed?
DISCUSSION
L290-292: please, clarify this sentence. You said that carcass yield of intensively reared pig was higher because. In which way would the higher feed intake compensate the higher weight of the digestive system of FR?
L306-307: To which parameter do you refer? I suppose that is IMF, but I suggest saying that LP and FR are similar, and differences are between both and SP, not between only LP and SP. Also, since you are discussing Table 2, actually only SV showed differences between different feeding regimens, I would underline this also at the beginning of the paragraph and not only in the last sentences (L318-320).
L319-320: I would suggest changing “more marked” since in LTL there are not significant differences at all
L323: Differences are still only in SV
L348-349: The results in SV profile of FR pigs are not unexpected, but the differences between LP and the other two treatments might require a deeper evaluation. Indeed, the following sentences are consistent with FR results but not with SP, being the group with higher protein level
